



# Distribution of chlorine and fluorine in benthic foraminifera

Anne Roepert[1], Lubos Polerecky[1], Esmee Geerken[2], Gert-Jan Reichart[1,2], and Jack J. Middelburg[1]

[1]Department of Earth Sciences, Utrecht University, PO Box 80021, 3508 TA Utrecht, The Netherlands
[2]Department of Ocean Systems, NIOZ Royal Netherlands Institute for Sea Research, and Utrecht University, 1790 AB
Den Burg, The Netherlands

**Correspondence:** Anne Roepert (a.roepert@uu.nl)

**Abstract.** Over the last decades a suite of inorganic proxies based on foraminiferal calcite have been developed, of which some
are now widely used for paleoenvironmental reconstructions. Studies of foraminiferal shell chemistry have largely focused on
cations and oxyanions, while much less is known about the incorporation of anions. The halogens fluoride and chloride are
conservative in the ocean, which makes them candidates for reconstructing paleoceanographic parameters. However, their po-
tential as a paleoproxy has hardly been explored, and fundamental insight in their incorporation is required. Here we used
nano-scale secondary ion mass spectrometry (NanoSIMS) to investigate, for the first time, the distribution of Cl and F within
shell walls of four benthic species of foraminifera. In the rotaliid species *Ammonia tepida* and *Amphistegina lessonii* Cl and F
were highly heterogeneous and correlated within the shell walls, forming bands that were co-located with the banded distribu-
tion of phosphorus. In the miliolid species *Sorites marginalis* and *Archaias angulatus* the distribution of Cl and F was much
more homogeneous without discernible bands. In these species Cl and P were correlated, whereas no correlation was observed
between Cl and F or between F and P. Additionally, their F content was about an order of magnitude higher than in the rotaliid
species. The high variance in the Cl and F content in the studied foraminifera could not be attributed to environmental parame-
ters. Based on these findings we suggest that in the rotaliid species Cl and F are predominately associated with organic linings.
We further propose that in the miliolid species Cl may be incorporated as a solid solution of chlorapatite or associated with
organic molecules in the calcite. The high F content together with the lack of correlation between Cl and F or P in the miliolid
foraminifera suggests a fundamentally different incorporation mechanism. Overall, our data clearly show that the calcification
pathway employed by the studied foraminifera governs the incorporation and distribution of Cl, F, P and other elements in their
calcite shells.

## 1 Introduction

Foraminifera are widely used to reconstruct paleoenvironments and climates based on relative species abundances and the
chemical or isotopic composition of their shells. Apart from the carbon and oxygen isotopic composition, most inorganic
proxies based on foraminiferal calcite involve cations or their isotopes (Boyle, 1981; Elderfield et al., 1996; Lea et al., 1999;
Allen et al., 2016), which can substitute for calcium in the calcite lattice. While there are also proxies based on the incorporation
of oxyanions such as iodate, sulphate and borate, and their isotopes (Lu et al., 2010; Paris et al., 2014; Yu et al., 2007; Rae





et al., 2011), much less is known about the incorporation of anions such as the halogens Cl, F and Br that are less likely to substitute for the carbonate ion (Kitano et al., 1975; Okumura et al., 1983, 1986).

The halogen anions are conservative in the ocean, which makes them potential candidates for reconstructing paleoceanographic parameters. Chlorine as a major constituent of sea salt is the most abundant halogen in seawater, followed by bromine, while fluorine concentrations are in the ppm range (e.g., Kendrick, 2018, and references therein). However, even though $Cl^-$ is

the most abundant (an)ion in seawater, the Cl content of inorganic and biogenic calcites is low as the $Cl^-$ anion does not fit well into the calcite lattice (Tokuyama et al., 1972; Kitano et al., 1975). In contrast, the $F^-$ anion has a much higher compatibility in many minerals compared to the heavier halogens, which can be attributed to its much smaller ionic radius (Kendrick, 2018). While seawater is enriched with the halogens chlorine and bromine with respect to the primary mantle (Kendrick, 2018), fluorine is strongly depleted in seawater with calcium carbonates being the major sink of dissolved fluoride in the oceans

(Carpenter, 1969).

Marine carbonates have F/Ca ratios ranging between $0.1$–$3.5\,\mathrm{mmol\,mol^{-1}}$ and $4.0$–$6.5\,\mathrm{mmol\,mol^{-1}}$ in calcite and aragonite, respectively (Carpenter, 1969), while Cl/Ca ratios range between $9$–$18\,\mathrm{\mu mol\,mol^{-1}}$ and $53$–$71\,\mathrm{\mu mol\,mol^{-1}}$ in inorganically precipitated calcite and aragonite, respectively (Kitano et al., 1975). The fluorine content of planktic foraminifera co-varies not only with $\delta^{18}O$ (Rosenthal and Boyle, 1993; Opdyke et al., 1993; Rosenthal et al., 1997), but also with Mg and Sr content

(Opdyke et al., 1993), and varies mostly in response to species-specific biological factors (Rosenthal and Boyle, 1993; Opdyke et al., 1993; Rosenthal et al., 1997). Furthermore, foraminiferal F content appears to be highly susceptible to diagenesis. Indeed, the release of F to porewater has been shown to be a proxy for carbonate dissolution during early diagenesis (Rude and Aller, 1991). The chlorine content of biogenic calcite has been proposed as a potential direct salinity proxy (Wit et al., 2013), but there are very few data on Cl in foraminifera (Szafranek and Erez, 1993; Erez, 2003).

Not much is known about how these anions are incorporated into foraminiferal shells. Possible incorporation modes include lattice-bound, interstitial, solid solution (e.g. fluorite, apatite), or bound to organic templates within the foraminiferal shell walls. Possible environmental factors playing a role during the incorporation processes of Cl and F may be related to pH homeostasis in the calcification fluid as $Cl^-$ and $F^-$ are the dissociated compounds of a strong and weak acid, respectively. Tanaka et al. (2013) showed that fluorine incorporation in non-symbiotic corals was governed by the carbonate ion concentration in

solution because of ion-exchange between fluoride and carbonate. A similar link has been suggested between foraminiferal fluorine content and carbonate ion concentration (Rosenthal and Boyle, 1993; Opdyke et al., 1993) as well as the presence of symbionts (Opdyke et al., 1993).

Furthermore, the spatial distribution of incorporated elements within foraminiferal shells can shed light on the incorporation mechanism. To the best of our knowledge, the spatial distribution of F has not been investigated yet, while Cl was previously

found at the location of organic linings in shell walls of *A. lobifera* (Szafranek and Erez, 1993; Erez, 2003). Recently, also the iodine micro-distribution in shell walls of *U. striata* was found to spatially correlate with organics (Glock et al., 2019). This suggests that halogen incorporation may be regulated by foraminiferal biomineralization pathways, although biogenic carbonate formation is ultimately expected to be governed by the same controlling factors as inorganic carbonate formation. The biomineralization pathway has consequences for trace element incorporation, as reflected in the occurrence of bands of





higher and lower concentrations of cations (e.g. Mg, Sr, Ba, Na) in perforate hyaline foraminiferal shell walls (Eggins et al., 2004; Sadekov et al., 2005; Kunioka et al., 2006; Branson et al., 2015; Jonkers et al., 2016; Geerken et al., 2018, 2019). This banding has been attributed to chamber addition in perforate hyaline foraminifera: an additional high-concentration band appears each time a new calcite lamella is added to the shell, which also covers previous chambers (Nehrke et al., 2013). Similarly, spine and the spine attachment zone of planktic foraminifera are enriched in Na while depleted in Mg, and vice

verca, respectively (Branson et al., 2016; Mezger et al., 2019; Bonnin et al., 2019).

Here we investigated, for the first time, the Cl and F distribution within foraminiferal shell walls of four benthic species on a sub-micron scale with NanoSIMS. Two rotaliid benthic species (*Ammonia tepida* and *Amphistegina lessonii*) and two miliolid benthic species (*Sorites marginalis* and *Archaias angulatus*) were investigated to cover benthic foraminifera with fundamentally different biomineralization pathways. Furthermore, one species is symbiont-barren (*Ammonia tepida*), whereas

the other three species are bearing photosynthetic symbionts.

## 2  Materials and Methods

### 2.1  Specimen selection and basic characterization

Six specimens of four species (two rotaliid and two miliolid species, respectively) from culturing experiments were investigated in this study (Table 1). The use of laboratory-grown specimens has the advantage that their growth conditions were controlled

and that potential effects of diagenesis on their shell halogen chemistry can be excluded.

Parameters of the carbonate system in the culture media, including alkalinity, carbonate ion concentration and saturation state with respect to calcite, were calculated from the measured DIC concentration, pH, salinity and temperature (Table 1). This was done in R (R Core Team, 2018) using the package *seacarb* (Gattuso et al., 2018). The equilibrium constants for the carbonate system $K_1$ and $K_2$ were taken from Lueker et al. (2000). Because DIC was not measured in the culture of *A. lessonii*,

we first applied the salinity-alkalinity relationship for the North Atlantic (Lee et al., 2006) to estimate the culture medium alkalinity, based on which we then calculated the remaining carbonate system parameters.

Prior to sample preparation for NanoSIMS analyses, each specimen was characterized with respect to its Na, Mg, Sr and Ba content. This was done by LA-ICP-MS as previously described (Geerken et al., 2018; van Dijk et al., 2019). For each specimen El/Ca ratios were determined on two or more chambers and averaged (Table A1).

### 2.2  Sample preparation and SEM imaging

The selected specimens of *A. tepida*, *A. lessonii*, *S. marginalis* and *A. angulatus* were embedded in epoxy resin (Araldite 2020) under vacuum and subsequently polished to expose cross-sections perpendicular to the shell walls (see Geerken et al. (2019) for details). Although epoxy resins contain high amounts of chlorine, we are confident that the halogens measured in the foraminiferal calcite are not due to contamination with resin nor an artifact of redeposition during the sputtering process

(Roepert, 2019). The first polishing steps used wet grinding papers of decreasing coarseness (HERMES, WS Flex 18C,





230 mm, P 800 and 219 ATM, SiC wet grinding paper, grain 4000) followed by agglomerated alpha alumina powder (Struers AP-A powder, grain size 0.3 μm) and $SiO_2$ powder (Logitech SF1 Polishing Suspension, grain size 0.035 μm). The polished samples were sputter-coated with 20 nm of Au using a sputter coater (JEOL JFC-2300HR high resolution fine coater with JEOL FC-TM20 thickness controller), after which they were imaged using a JEOL Neoscope II JCM-6000 table-top SEM to
identify suitable areas for NanoSIMS analysis (Figure A1).

## 2.3 NanoSIMS imaging

NanoSIMS analysis was performed with the Cameca NanoSIMS 50L instrument available at Utrecht University. Using an element standard (SPI Supplies, 02757-AB 59 Metals & Minerals Standard), magnetic field and exact positions of the electron multiplier detectors were adjusted to enable detection of negative secondary ions $^{12}C^-$, $^{16}O^-$, $^{19}F^-$, $^{31}P^-$, $^{35}Cl^-$, $^{37}Cl^-$, and
$^{40}Ca^{16}O^-$. The secondary ions were sputtered from the sample surface using an 8 kV primary $Cs^+$ ion source.

Because the primary ion beam was positive, calcium had to be detected as $^{40}Ca^{16}O^-$ and not as $^{40}Ca^+$. However, because the Ca:O stoichiometry in calcite with trace amounts of organics is fixed, we assumed that the measured distribution of $^{40}Ca^{16}O$ well approximated the distribution of Ca. This assumption was supported by the good correlation between the secondary ions $^{40}Ca^{16}O^-$ and $^{16}O^-$ detected from the calcite (Figure A2). $^{31}P$ was measured as a tracer for organics in the calcite (Geerken
et al., 2019), whereas $^{12}C$ was measured to help distinguish between resin and calcite. To ensure that the detection of $^{35}Cl$ was not influenced by possible isobaric interferences such as $^{16}O^{18}O^1H$ and $^{34}S^1H$, we used sufficiently high mass resolution power (MRP > 5113) and additionally measured the isotope $^{37}Cl$ as well. The obtained $^{37}Cl/^{35}Cl$ ratio differed from the natural abundance ratio of 0.320 by no more than 0.015, confirming that the influence of isobaric interferences for the detection of Cl was negligible. Similarly, separation of $^{19}F$ from the possible interference by $^{18}O^1H$ was achieved by using MRP > 2214,
whereas interferences from molecules such as $^{12}C^7Li$, $^{13}C^6Li$ or $^{16}O^1H_3$ are highly unlikely.

Before each analysis the area of interest was pre-sputtered for 10–15 min until secondary ion counts stabilized. Subsequently, ion count images were acquired by rastering the primary beam (dwell time of 1 ms pixel$^{-1}$) over the sample surface using the diaphragm and slit settings listed in Table A2. The primary beam current on the sample surface ranged between 0.5–2 pA depending on the size of the imaged area. The spatial resolution ranged between 50–100 nm pixel$^{-1}$. All analyses employed
an e-gun to avoid charging of the sample surface. Because some of the secondary ion counts were very low, the imaged areas were measured multiple times (250–1000, depending on the sample) and the signals from the individual planes were aligned and accumulated.

## 2.4 Data processing

NanoSIMS data were processed with the freeware program Look@NanoSIMS (Polerecky et al., 2012). After alignment and
accumulation of the image data, regions of interest (ROIs) corresponding to foraminiferal calcite were drawn by hand based on the $^{40}Ca^{16}O^-$ image. With the additional use of the $^{12}C^-$ and $^{35}Cl^-$ images, areas of exposed resin or pores within the foraminiferal walls were identified and excluded from the final analysis.





Due to the lack of reliable calibration standards, and because Ca was measured as the molecular ion $^{40}Ca^{16}O^-$, the El/Ca ratios are reported in this study by the ratios between the raw data, i.e., the secondary ion counts El$^-$ and $^{40}Ca^{16}O^-$. Although

not fully quantitative, the ratios calculated in this way are comparable between the different foraminifera specimens and species because the secondary ion counts detected by NanoSIMS are linearly proportional to the concentration of the corresponding element, and because all the measurements were done in a similar biogenic calcite matrix using the same pre-sputtering and measuring protocol.

To ensure that the El/Ca ratios were not affected by insufficient pre-sputtering, depth profiles of the El$^-$/$^{40}Ca^{16}O^-$ ion count

ratios were inspected for each ROI, and the planes where the ratios showed a significant trend with depth were excluded from the final analysis. Lateral profiles of the El$^-$/$^{40}Ca^{16}O^-$ ion count ratios perpendicular to the shell surface were extracted from the accumulated NanoSIMS images. The width of the profile line, which corresponds to the amount of averaged lateral profiles, was set to $20\,\mathrm{pixel}$ to increase the signal-to-noise-ratio.

To investigate the spatial correlation of El$^-$/$^{40}Ca^{16}O^-$ ion count ratios, ROIs were drawn on the NanoSIMS images in

Look@NanoSIMS in such a way, that regions of higher and lower ion count ratios on the foraminifera were separated into different ROIs to separate the spatial variability. That way, 40 to 47 separate data points (ROIs) per species, grouped per specimen, were extracted from the NanoSIMS images. Subsequently, correlation matrices were calculated for the accumulated ion count ratios in those ROIs using the corrplot package (Wei and Simko, 2017) in R (R Core Team, 2018).

## 3 Results

### 3.1 Spatial distribution of chlorine and fluorine


The halogens Cl and F show distinct banding in the rotaliids, in particular in *A.tepida* (Figure 1). Moreover, in the rotaliids, maxima of Cl/Ca and F/Ca are co-located with those of P/Ca, and correlate well in *A. tepida*, while the correspondence between P/Ca and F/Ca is moderate in *A. lessonii*, and spatially rather complex for Cl/Ca (see lateral profiles in Figure 1). Furthermore, the contrast between the high-intensity and low-intensity bands in F/Ca, Cl/Ca and P/Ca is higher in *A. tepida* than in *A. lessonii*.

In the miliolid foraminifers no banding of neither halogens nor P is visible, with the exception of a slight elevation in Cl and P in areas of an image that were identified as a suture in SEM images. Lateral profiles in *A. angulatus* show a correlation of Cl with P, and no correlation of F with Cl or P. The lateral profile through the shell wall of *S. marginalis* shows similar patterns as the one of *A. angulatus*. The F/Ca ion count ratios in the miliolids *A. angulatus* and *S. marginalis* are in the same range and one order of magnitude higher than those in rotaliid *A. tepida* and *A. lessonii* (Figures 1 and 2). The Cl/Ca and P/Ca ion count

ratios are in the same order of magnitude in all four species (Figures 1 and 2).

Ion count ratios of F/Ca and Cl/Ca correlate with each other in *A. tepida*, while there is no correlation in *A. lessonii* and the miliolids (Figure 2A and A3). Both F/Ca and Cl/Ca are correlated with P/Ca in the rotaliids, while only Cl/Ca is correlated with P/Ca in the miliolids (Figure 2B,C). All correlations described here are significant to a level of $p < 0.001$; $R^2$ and p-values of the correlations are reported in Figure A3. These correlations are also seen when the elements F, Cl and P are normalized to O

instead being normalized to Ca (Figure A5).





## 3.2  Relation with cation incorporation and culture media properties

In all four species, the Cl content does not correlate to any of the elemental ratios measured by LA-ICP-MS (upper panels in Figure 3). However, the elevated F/Ca ratios in the miliolid foraminifera coincide with elevated Mg/Ca and Ba/Ca, which also are an order of magnitude higher in these species than in the rotaliid foraminifera (lower panels in Figure 3). Our data show no
correlation of Cl/Ca or F/Ca with neither Na/Ca nor Sr/Ca (Figure 3).

Our data show no correlation of Cl/Ca or F/Ca with salinity or temperature (Figure A4). Furthermore, crossplots of the NanoSIMS Cl/Ca and F/Ca ion count ratios show no correlation with carbonate system parameters for Cl/Ca (Figure 4). However, NanoSIMS F/Ca ion count ratios are higher in the miliolid foraminifera, which were cultured at higher DIC, corresponding to higher alkalinity and carbonate ion concentration as well (Figure 4).

## 4  Discussion

### 4.1  Limited environmental control on Cl and F incorporation into
foraminiferal shells

The NanoSIMS data presented clearly show that biomineralization pathways govern the incorporation and distribution of Cl and F within foraminiferal shells: the rotaliid species show distinct banding in Cl, F and P, while the F-rich miliolid species
do not. Biologic control is known to affect incorporation of most elements into foraminiferal shells, while at the same time relationships with environmental conditions have proven robust tools for paleo reconstructions (Eggins et al., 2004; Kunioka et al., 2006; Paris et al., 2014; Spero et al., 2015; Fehrenbacher et al., 2017; Geerken et al., 2019). In our data set comprised of a very limited amount of specimens, we see no overall trend in Cl/Ca ratios in foraminiferal calcite with chemical properties of the culture media. The high intra-shell variability in rotaliid foraminifera and the spatial correlation with P on the location of
organic linings suggest that Cl is associated with organic linings in rotaliid foraminiferal shells. Furthermore, Cl/Ca is highly variable within a single section through a foraminiferal wall in all the species measured and the range of Cl/Ca ratios is similar in all investigated specimens.

The overall absence of trends with environmental conditions as well as the high intra-specimen variability lowers the confidence in potentially using Cl/Ca for paleo reconstructions. However, as this study does not cover a range of physicochemical
parameters for a single species, but rather presents a collection of different species that were also grown in different conditions, we cannot exclude that any of the presented physicochemical parameters may exert an influence on the Cl content of foraminifera on a species-specific level. A definite conclusion regarding proxy applicability would require culturing studies including 20–30 specimens per species per environmental condition.

Moreover, there is no discernable trend of Cl/Ca ratios in foraminiferal calcite with any of the measured trace elements
(Figure 3). Cl/Ca ratios are in the same range for species with low-Mg calcite (rotaliid) as they are for those with high-Mg calcite (miliolid), suggesting that chlorine incorporation is systematically different from the incorporation of these cations. In inorganically precipitated calcite, chlorine contents are an order of magnitude lower than sodium contents (Kitano et al.,





1975), suggesting that Cl is incorporated neither as fluid inclusions (Wit et al., 2013) nor as solid solutions of NaCl into calcite. As chlorine content seems not to reflect any environmental parameter, and Cl/Ca correlates well with P/Ca in all the species

investigated here, we suspect that chlorine incorporation into foraminiferal calcite is closely related to organic molecules involved in calcification or to a solid-solution between calcite and chlorapatite ($Ca_5(PO_4)_3(OH,Cl)$).

The F/Ca ratios in the miliolid species are about an order of magnitude higher than those in the rotaliid species. The elevated F/Ca ratios in miliolids coincide with overall higher $CO_3^{2-}$-ion concentration in the culture media of the miliolid species. This might indicate a relation between foraminiferal F/Ca ratios and carbonate ions, but the relationship is inverse to what one

would expect for inorganic ion exchange (Ichikuni, 1979) and what has been observed in corals (Tanaka et al., 2013). However, the high intra-shell variability in F/Ca ratios of single specimens and the co-variation of environmental conditions with mineralization pathway complicates attributing F/Ca ratios to environmental parameters. Species-specific culturing studies could provide more insight into whether F/Ca ratios of benthic foraminifera on a species-specific level correlate with environmental conditions, as then the effect of different biomineralization pathways would not hamper interpretation as is the case in our data

set.

Opdyke et al. (1993) suggested that the presence of photosynthetic symbionts in foraminifera impacts their F/Ca ratio. Symbionts influence the intracellular carbonate chemistry by photosynthesis, which could link to fluoride incorporation via the intracellular $CO_3^{2-}$ ion activity. *A. tepida* is the only symbiont-barren species we investigated, and indeed its F/Ca ratios are lower than in the miliolid symbiont-bearing species. However, the symbiont-bearing rotaliid *A. lessonii* exhibits the lowest

mean F/Ca ratios, which are in the same order of magnitude as in *A. tepida*, but less variable throughout the shell wall. We therefore conclude that F/Ca ratio is unlikely directly related to the presence of symbionts in foraminifera.

Notably, F/Ca ratios are higher in specimens with a higher Mg and Ba content (Figure 3). A correlation of trace element content with Mg content within and between species has been found for several elements, including Sr, Zn, and Na (Evans et al., 2015; van Dijk et al., 2017; Geerken et al., 2018), and also F (Opdyke et al., 1993). The miliolid species have generally

a much higher Mg content than the rotaliid species and their biomineralization mechanisms are thought to be substantially different (Figure 5). The fact that higher F content corresponds with higher Mg content may point towards a strong biological control on F incorporation. Fluorine may be incorporated in solid solutions. Fluorite ($CaF_2$) solid solution has been suggested as the incorporation mechanism for fluorine in calcite (Carpenter, 1969), but also fluorapatite ($Ca_5(PO_4)_3(OH,F)$) solid solution would be a possible option.

## 4.2 Cl and F incorporation in foraminifera is primarily controlled by biomineralisation pathway

In the two species of rotaliid foraminifera that were investigated here, Cl and F show strong banding. The Cl and F bands are co-located with P in the foraminiferal shell walls. Since phosphorus is present in organic molecules like phospholipids in membranes, P/Ca can be used to image organic linings in between the lamella in hyaline foraminiferal shell walls, as demonstrated in Geerken et al. (2019). In *A. tepida*, the correlation of Cl and F with P is tight, and we conclude that both

elements are primarily associated with the organic linings. In *A. lessonii*, the peaks in Cl/Ca and F/Ca lateral profiles are also co-located with peaks in P/Ca. However, in the specimens we analyzed, there seems to be substantial additional Cl and F also





in some calcite lamella, as can be seen in the lateral profiles. Moreover, the contrast between high-intensity bands and low-intensity bands is less prominent in *A. lessonii* than in *A. tepida*. We suggest that also in *A. lessonii*, association with organic linings is the primary mode of incorporation of both elements in the foraminiferal shells. Using NanoSIMS and the very same

species and specimens, Geerken et al. (2019) reported co-occurrence of organic linings and banding of Mg, Na, Sr, K, S, P and N. Moreover, they showed that elemental incorporation in *A. lessonii* was overall higher than in *A. tepida*, consistent with our observations for the halogens (Figures 1, 2).

In the miliolids, the distributions of Cl and F are distinctly different from those of the rotaliids: since miliolids do not calicify by adding subsequent lamella of calcite (Figure 5), no patterns of alternating high and low concentration banding are

visible. Moreover, Cl and F are spatially not correlated throughout the shell walls of these miliolids, indicating that Cl and F may have different modes of incorporation. The correlation of Cl and P within the shell wall supports the hypothesis that Cl is incorporated in the calcite as a solid solution of chlorapatite, a mineral containing both Cl and P, or associated with organic molecules as for the rotaliids, but then distributed in a less organized way (no banding) within the calcite. The type of organics being present in foraminiferal calcite is determined by the precipitation pathway: rotaliids initiate calcification around

a primary organic sheet (POS) and cover their shell with organic linings, while miliolid shells comprise of randomly oriented calcite needles held together in an organic matrix. As these organics may differ in their P content, it is possible to measure comparable P contents in both rotaliid and miliolid calcite, even though their absolute organic matter content within the calcite is expected to be different. Similar Cl content in both rotaliids and miliods may thus be due to differences in the composition of the organics or may hint to an incorporation via a different pathway. Alternatively, apatite may form via the adsorption

of phosphate to calcite and amorphous calcium phosphates at low Mg concentrations in solution (Martens and Harriss, 1970; Millero et al., 2001). The incorporation of Cl via chlorapatite could also explain the spatial correlation of Cl and P in the species we analysed. If F would be incorporated similarly as Cl, we would expect the F and Cl distribution to be comparable. As this is not the case, we conclude that F is incorporated primarily in a different way than Cl, e.g. as a solid solution of fluorite, as suggested before (Rosenthal and Boyle, 1993).

The observed difference between F-rich miliolids lacking organic linings and clear banding of trace elements on the one hand, and the rotaliids with F, Cl and P rich bands on the other hand, is consistent with known differences in calcification mechanisms (Figure 5) that have developed during the evolution of foraminifera (Debenay et al., 1996; Bentov and Erez, 2006). Hyaline (including the rotaliid) foraminifera precipitate calcite onto organic templates within an extracellular but confined space, and add a new lamella to the entire shell each time they produce a new chamber (Hemleben et al., 1986; de Nooijer

et al., 2014). For some intermediate Mg-calcite producing foraminifera (like *A. lobifera* or also *A. lessonii*), transport of vesicles to the site of calcification has been observed suggesting controlled biomineralization (at least partly) from internal Ca and carbonate pools (e.g., de Nooijer et al., 2009). Furthermore, in hyaline foraminifera, selective ion transport to the site of calcification via trans-membrane pumping of elements is discussed as the biological control on biomineralization (Nehrke et al., 2013; de Nooijer et al., 2014; Toyofuku et al., 2017). In contrast, porcelaneous miliolid foraminifera produce a wall

of high-Mg calcite with thin inner and outer layers connected by a thick middle layer of crystal needles (Hemleben et al., 1986; Debenay et al., 1996), which are glued together by an organic matrix. Miliolid foraminifera are generally thought to

precipitate calcium crystals in intracellular vesicles prior to arranging them in the shape of the new chamber wall (Angell, 1980; Hemleben et al., 1986; Debenay et al., 1998, 2000; Bentov and Erez, 2006). Furthermore, there are species within the suborder of the miliolids that show features of both biomineralization pathways, such as *Archaias angulatus*, which appears to

precipitate calcite at the site of the new chamber wall, opposed to other miliolid species (Wetmore, 1999).

During calcification, miliolids enclose seawater vesicles (Figure 5) and then produce calcite rich in various cations (van Dijk et al. (2017), Table A1) and F (Figure 1). The rotaliids such as *A. tepida* may use highly selective ion channels and organic layers to deposit new calcite. As a consequence, these calcites are low in Mg, Ba and the halogens (F, Cl) and show distinct banding. The rotaliid *A. lessonii* produces calcite with intermediate Mg contents and less distinct banding for cations (Geerken

et al., 2019), indicative of less biological control on ion transport and calcite deposition than in *A. tepida*. Our results for P, Cl and F support this.

## 5 Conclusions

Here we investigated for the first time the spatial distribution of the halogens Cl and F in foraminiferal shell walls. In the rotaliid benthic species *Ammonia tepida* and *Amphistegina lessonii*, Cl and F are distributed in bands within the chamber walls,

which co-locate with P banding. In the miliolid benthic species *Sorites marginalis* and *Archaias angulatus* Cl and F were not found to occur in bands. However, the rather homogeneously-distributed Cl was found to correlate with P content, while F did not correlate with either P or Cl. Based on these findings we suggest that Cl and F are predominately associated with organic linings in the rotaliid species. We further propose that Cl may be incorporated in miliolid species as a solid solution of chlorapatite or be associated with organics. Our data in the miliolid species suggests that F is incorporated in a different way

than Cl, as F does not correlate with P.

*Data availability.* The data presented in this study are available at doi:10.4121/uuid:9951e801-5574-498e-b375-fa6941a0f071.

*Author contributions.* JJM, GJR and AR designed the experiments. EG prepared the samples for NanoSIMS analyses. EG and AR conducted the SEM analyses. AR conducted the NanoSIMS analyses. AR and LP performed the data analysis. AR interpreted the data, prepared the figures and wrote the manuscript text with contributions from all authors.

*Competing interests.* The authors declare no competing interests.

*Acknowledgements.* We thank Inge van Dijk and Lennart de Nooijer for providing *A. lessonii* specimens, and Michiel Kienhuis for analytical support. The NanoSIMS facility at Utrecht University was financed through a large infrastructure grant by the Netherlands Organisation for





Scientific Research (NWO) (grant no. 175.010.2009.011). This work was carried out under the programme of the Netherlands Earth System Science Centre (NESSC), financially supported by the Ministry of Education, Culture and Science (OCW) (grant no. 024.002.001).





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





**Table 1.** Studied foraminifera specimens and the corresponding culture conditions.

| # | species[1] | salinity | T [°C] | DIC [μmol/kg] | alkalinity [μmol/kg] | pH | $CO_3^{2-}$ [μmol/kg] | $\Omega_{\text{calcite}}$ |
|---|---|---|---|---|---|---|---|---|
| 1[†] | *A. tepida* (R) | 25.2 | 25 | 1087 | 1350 | 8.32 | 162 | 4.22 |
| 2[‡] | *A. lessonii* (R) | 35.2 | 21.2 | 2069 | 2314 | 8.00 | 177 | 4.24 |
| 3[*] | *A. angulatus* (M) | 40 | 25 | 3861 | 4477 | 8.10 | 506 | 11.67 |
| 4[*] | *A. angulatus* (M) | 30 | 25 | 2644 | 3153 | 8.27 | 399 | 10.00 |
| 5[*] | *S. marginalis* (M) | 30 | 25 | 2644 | 3153 | 8.27 | 399 | 10.00 |
| 6[*] | *S. marginalis* (M) | 40 | 25 | 3861 | 4477 | 8.10 | 506 | 11.67 |

[1]  R = rotaliid, M = miliolid.

[†]  Selected from the culture experiment by Geerken et al. (2018).

[‡]  Selected from the culture experiment by van Dijk et al. (2019).

[*]  Selected from an unpublished culture experiment (see Appendix B).





**Figure 1.** Spatial distribution of Cl/Ca, F/Ca, and P/Ca ratios in the calcite shells of the studied foraminifera species. Shown are representative images as well as lateral profiles along a line going from the inside to the outside of the shell as depicted by an arrow in the images. Note that the displayed ratio images are log-transformed, and that the color-scale for the given ratio is the same for all species. Blacked out areas correspond to resin. The scale bar in each image is 5 μm.



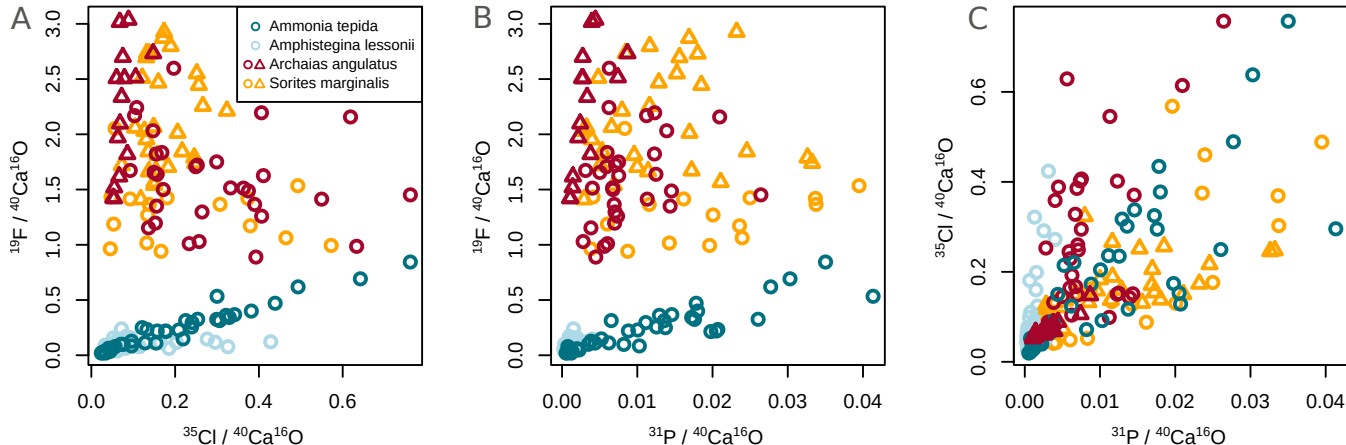

**Figure 2.** Variability of the different El/Ca ratios within shells of the studied foraminifera. Individual data-points represent multiple ROIs drawn on the images as described in Section 2.4. Symbol shapes depict different specimens.



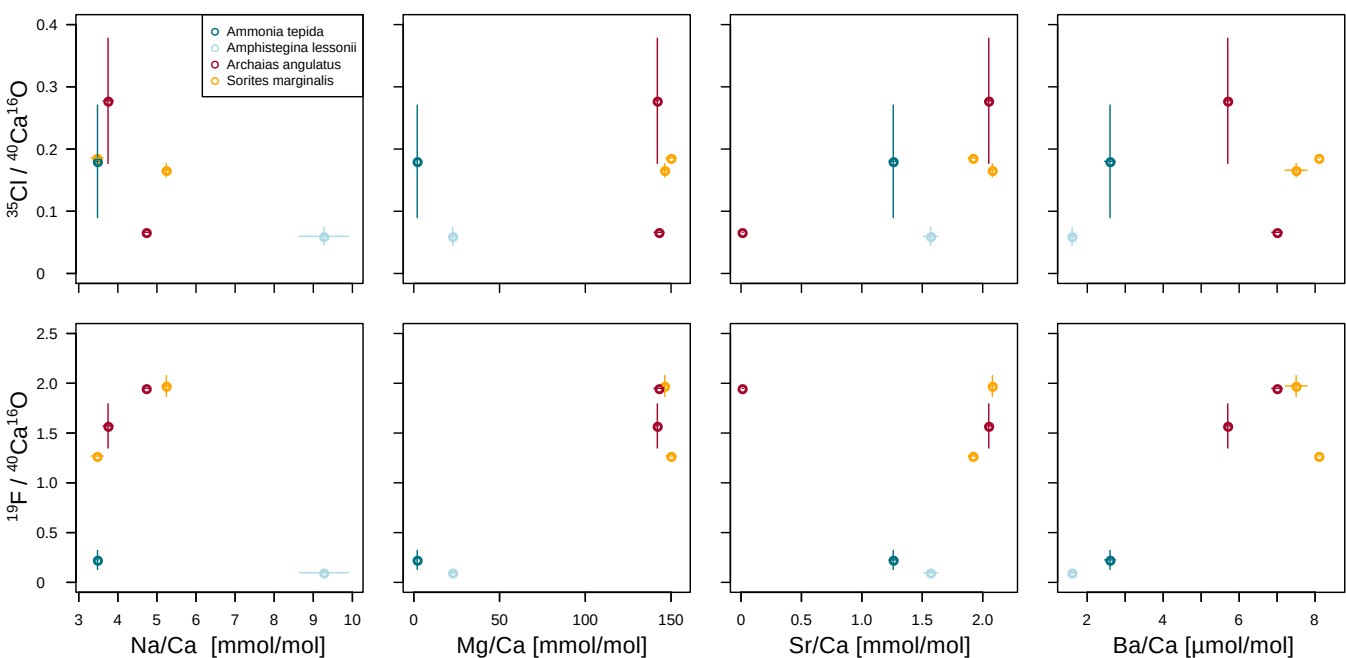

**Figure 3.** Crossplots of NanoSIMS $^{35}Cl/^{40}Ca^{16}O$ and $^{19}F/^{40}Ca^{16}O$ ion count ratios with LA-ICP-MS El/Ca ratios in the same specimen. The error bars depict one standard error of the mean NanoSIMS ion count ratios where more than one image was analysed per specimen, and one standard error of duplicate or triplicate LA-ICP-MS measurements.



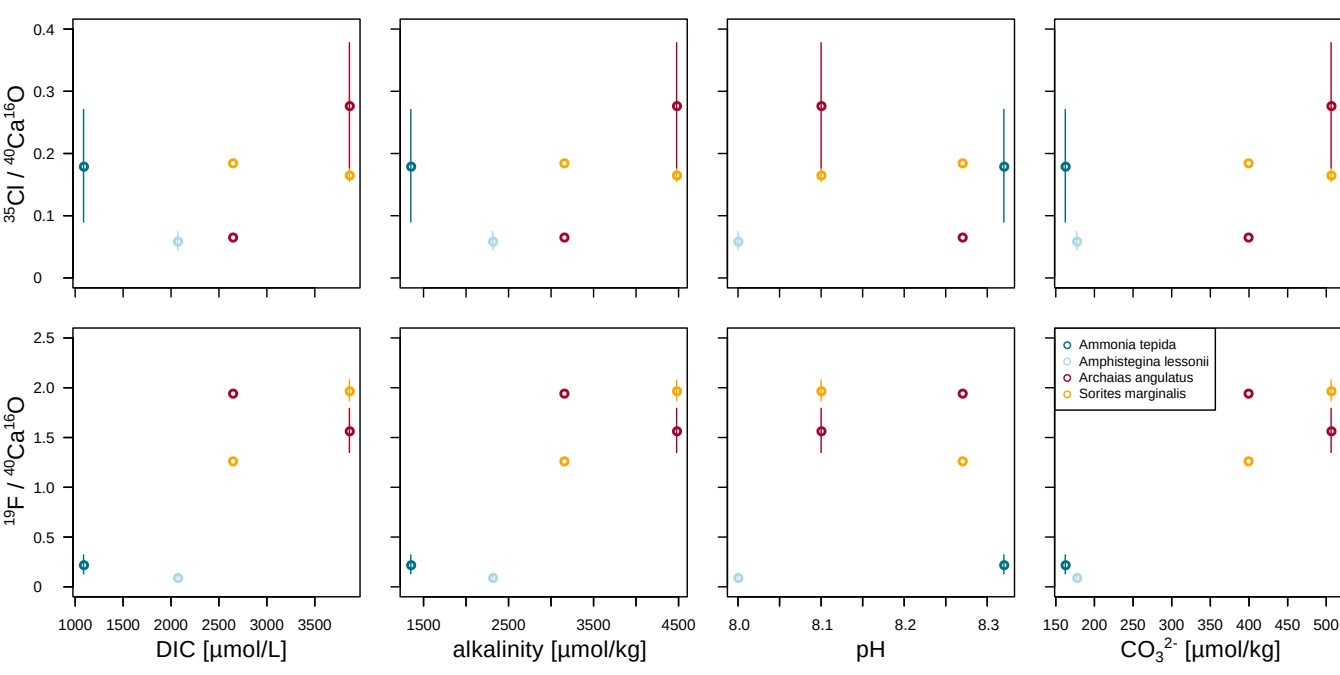

**Figure 4.** Crossplots of NanoSIMS $^{35}Cl/^{40}Ca^{16}O$ and $^{19}F/^{40}Ca^{16}O$ ion count ratios with carbonate system parameters of the culture media. The error bars depict one standard error of the mean NanoSIMS ion count ratios where more than one image was analysed per specimen.

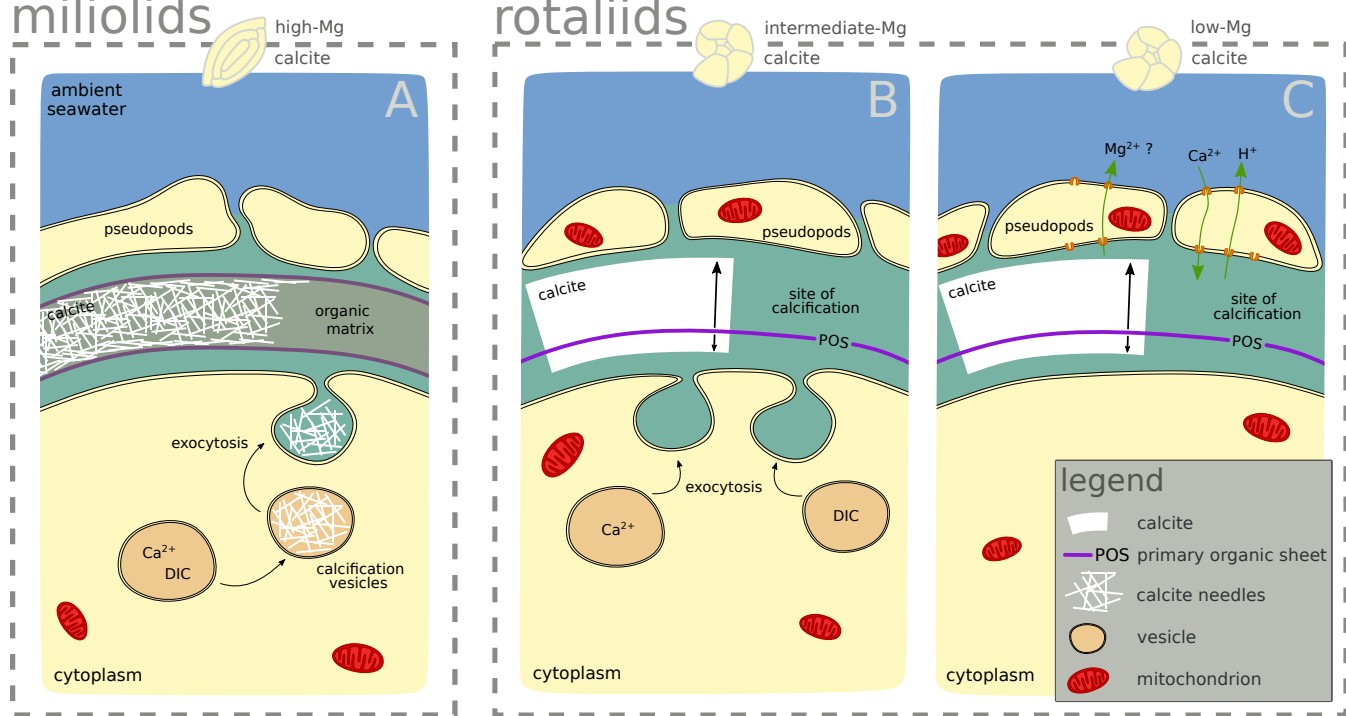

**Figure 5.** Scheme highlighting differences of the calcification mechanisms in miliolid and rotaliid foraminifera. Miliolid foraminifera precipitate calcium crystals in intracellular vesicles prior to arranging them in the shape of the new chamber wall (A). Rotaliid foraminifera precipitate calcite onto organic templates within an extracellular but confined space, and add a new lamella to the entire shell each time they produce a new chamber (B, C). In intermediate Mg-calcite producing rotaliid foraminifera (like *A. lessonii*), transport of vesicles to the site of calcification has been observed suggesting controlled biomineralization (at least partly) from internal Ca and carbonate pools (B). In rotaliid low Mg-calcite producing foraminifera, selective ion transport to the site of calcification via trans-membrane pumping of elements is discussed as the biological control on biomineralization (C).





**Appendix A: Supplementary tables and figures**

**Figure A1.** SEM images of the analysed specimens with the areas imaged by NanoSIMS indicated by white or yellow squares: *Ammonia tepida* (#1), *Amphistegina lessonii* (#2), *Archaias angulatus* (#3, #4), *Sorites marginalis* (#5, #6). The specimen numbers correspond to those in Table 1 and A1. The yellow squares indicate the locations of the NanoSIMS images shown in Figure 1. SEM images are flipped horizontally to facilitate navigation in the NanoSIMS instrument, where the secondary ion images are horizontally mirrored.



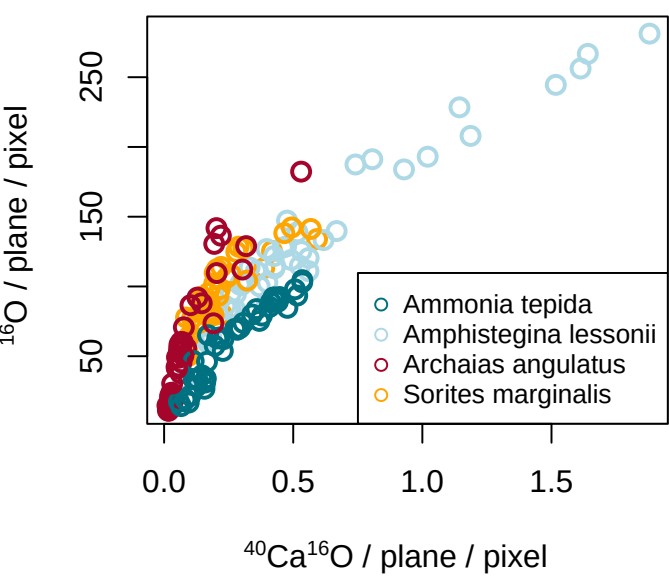

**Figure A2.** NanoSIMS $^{40}$Ca$^{16}$O/plane/pixel and $^{16}$O/plane/pixel in four foraminifera species. The NanoSIMS ion count rates of 40Ca160 and $^{16}$O highly correlate in calcite (Figure A2), which is why elemental ratios can be normalized to both for display. Since elemental ratios determined in bulk calcite are normalized to calcium, we present the NanoSIMS data mostly as El/$^{40}$Ca$^{16}$O ratios.





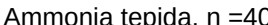

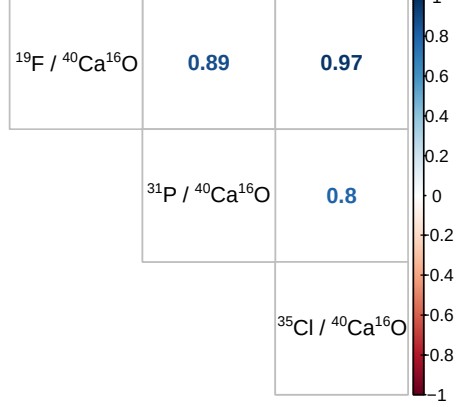
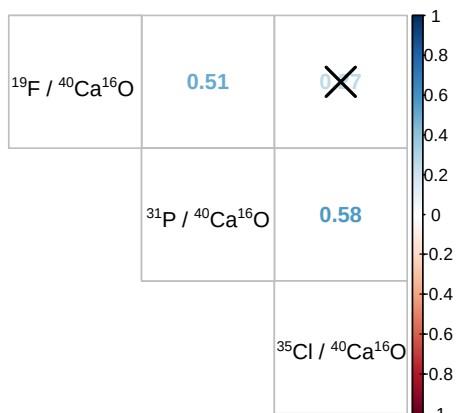

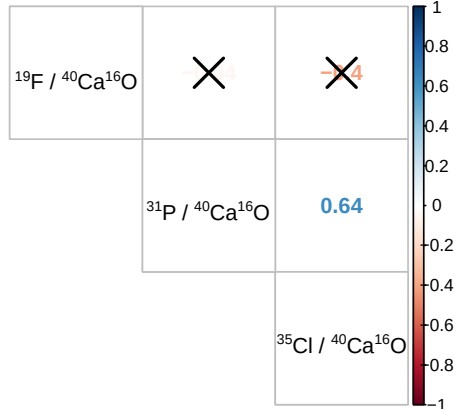
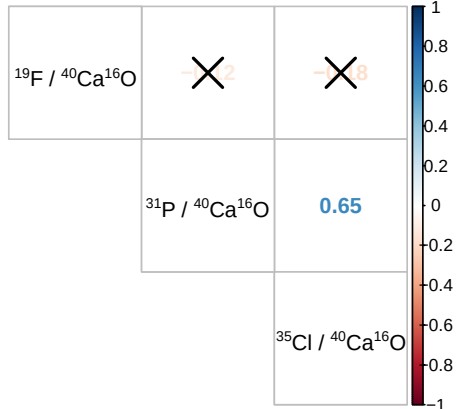

**Figure A3.** Correlation matrices of NanoSIMS $^{19}F/^{40}Ca^{16}O$, $^{31}P/^{40}Ca^{16}O$, and $^{35}Cl/^{40}Ca^{16}O$ ion count ratios per foraminiferal species across n ROIs. Coloured values in the matrix are $R^2$ and the correlation is significant to a level of $p < 0.001$ wherever no black cross present. The plot was created using the corrplot package in R (Wei and Simko, 2017; R Core Team, 2018).

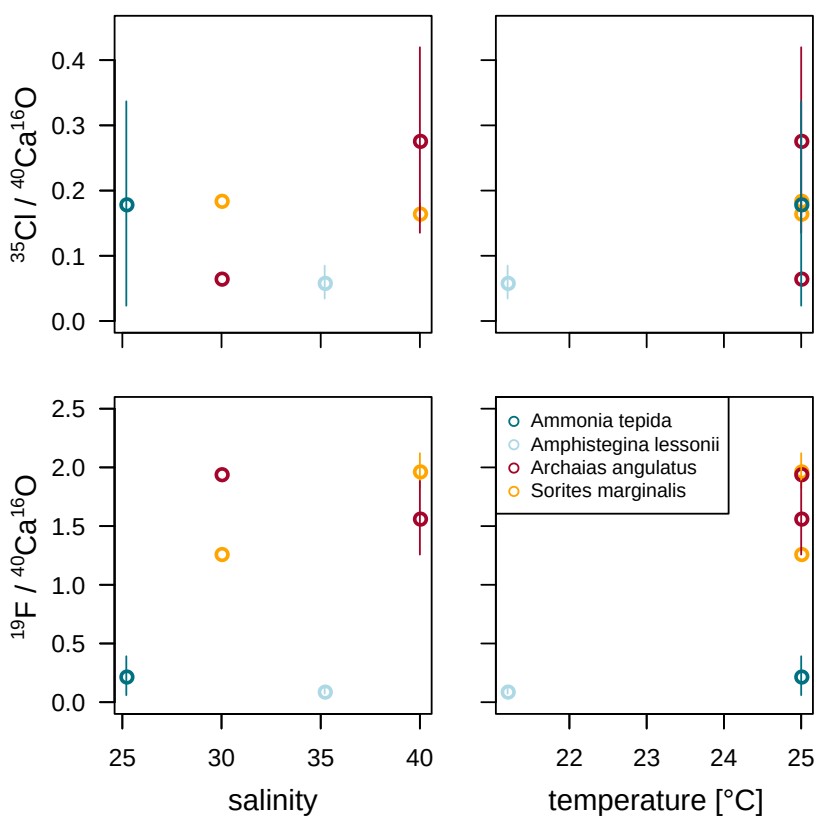

**Figure A4.** NanoSIMS $^{35}$Cl/$^{40}$Ca$^{16}$O and $^{19}$F/$^{40}$Ca$^{16}$O ion count ratios in the measured specimen versus temperature and salinity of the culture media. The error bars depict the standard deviation of the mean NanoSIMS ion count ratios for those specimen where more than one image was analysed on the shell.





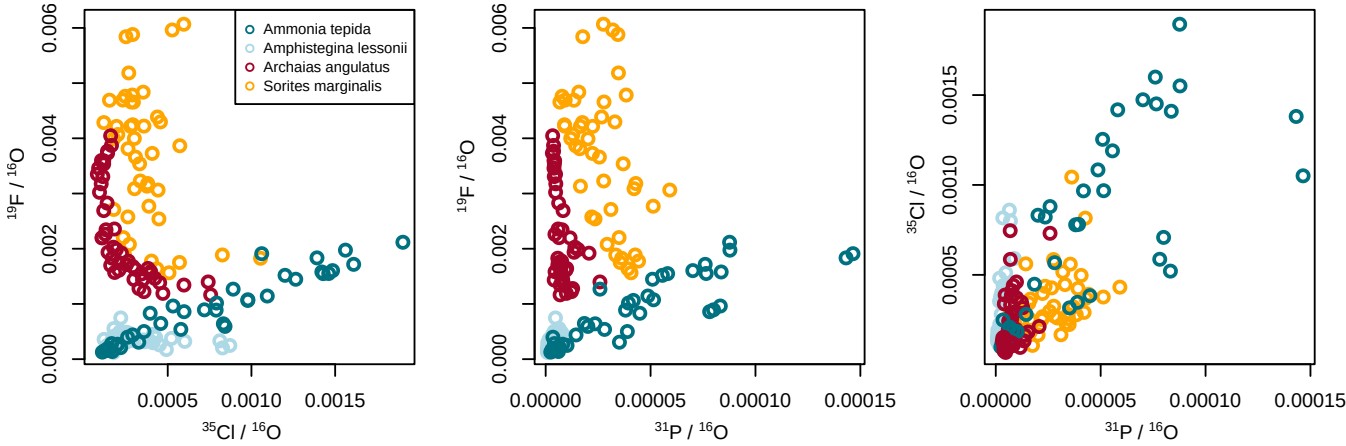

**Figure A5.** Illustration to show that the patterns in figure 2 do not change much when normalizing to $^{16}$O instead of $^{40}$Ca$^{16}$O.





**Table A1.** Additional information of specimens. LA-ICP-MS data of average single shell El/Ca ratios were determined on several chambers.

| # | specimen ID* | species | symbionts | Na/Ca [mmol/mol] | Mg/Ca [mmol/mol] | Sr/Ca [mmol/mol] | Al/Ca [mmol/mol] | Ba/Ca [mmol/mol] |
|---|---|---|---|---|---|---|---|---|
| 1 | 34_61 | Ammonia tepida | no | 3.48 ± 0.03 | 2.0 ± 0.1 | 1.26 ± 0.05 | 0.009± < 0.000 | 0.0026 ± 0.0002 |
| 2 | 13_110 | Amphistegina lessonii | yes | 9.27 ± 1.10 | 22.7 ± 3.8 | 1.57 ± 0.10 | 0.008± < 0.000 | 0.0016 ± 0.0001 |
| 3 | 32_89 | Archaias angulatus | yes | 3.75 ± 0.22 | 142.0 ± 1.0 | 2.05 ± 0.06 | 0.011± < 0.000 | 0.0057 ± 0.0001 |
| 4 | 46_87 | Archaias angulatus | yes | 4.73 ± 0.22 | 143.0 ± 4.2 | 0.01± < 0.00 | 0.008 ± 0.001 | 0.0070 ± 0.0002 |
| 5 | 01_87 | Sorites marginalis | yes | 3.47 ± 0.27 | 150.0 ± 5.2 | 1.92 ± 0.07 | 0.153 ± 0.093 | 0.0081± < 0.000 |
| 6 | 30_89 | Sorites marginalis | yes | 5.24 ± 0.19 | 146.3 ± 2.5 | 2.08 ± 0.02 | 0.044 ± 0.013 | 0.0075 ± 0.0005 |

* Lab internal specimen ID based on stub#_number. For *A. tepida* as in Geerken et al. (2018).





**Table A2.** NanoSIMS 50L acquisition settings.

| NanoSIMS settings[*] | *Ammonia tepida* | *Amphistegina lessonii* | *Archaias angulatus* | *Sorites marginalis* |
|---|---|---|---|---|
| **Pre-sputtering** | | | | |
| beam current (in $FC_o$) [pA] | $\sim 140$ (preset 10 in D1–3) | $\sim 280$ (preset 20 in D1–3) | $\sim 280$ (preset 20 in D1–3) | $\sim 140$–280 (preset 10–20 in D1–3) |
| diaphragm and slits | D1–1 | D1–1 | D1–1 | D1–1 |
| FOV [μm$^2$] | $8 \times 8$ | $12 \times 12$ to $23 \times 23$ | $55 \times 55$ | $25 \times 25$ to $40 \times 40$ |
| time [min] | 10 | 10 | 15 | 10–15 |
| eGun | on | on | on | on |
| **Image Aquisition** | | | | |
| beam current (in $FC_o$) [pA] | 0.5 | 1 | 2 | 1 |
| diaphragm and slits | D1–3, ES–3, AS–2, EnS–1 | D1–3, ES–3, AS–2, EnS–1 | D1–3, ES–3, AS–2, EnS–1 | D1–3, ES–3, AS–2, EnS–1 |
| Detected masses | $^{12}$C, $^{16}$O, $^{19}$F, $^{31}$P, $^{35}$Cl, $^{37}$Cl, $^{40}$Ca$^{16}$O | $^{12}$C, $^{16}$O, $^{19}$F, $^{31}$P, $^{35}$Cl, $^{37}$Cl, $^{40}$Ca$^{16}$O | $^{12}$C, $^{16}$O, $^{19}$F, $^{31}$P, $^{35}$Cl, $^{37}$Cl, $^{40}$Ca$^{16}$O | $^{12}$C, $^{16}$O, $^{19}$F, $^{31}$P, $^{35}$Cl, $^{37}$Cl, $^{40}$Ca$^{16}$O |
| dwell time [μs pixel$^{-1}$] | 1000 | 1000 | 1000 | 1000 |
| image FOV [μm$^2$] | $6 \times 6$ | $8 \times 8$ to $20 \times 20$ | $50 \times 50$ | $23 \times 23$ to $30 \times 30$ |
| image resolution [px×px] | $128 \times 128$ | $128 \times 128$, once $256 \times 256$ | $256 \times 256$ | $256 \times 256$ |
| number of planes | 1000 | 1000, once 250 | 250 | 250 |
| eGun | on | on | on | on |

[*] Abbreviations: objective current ($FC_o$); field of view (FOV); electron flood gun (eGun).

## Appendix B: Culture conditions of the foraminiferal species

The *A. angulatus* and *S. marginalis* specimens were collected in Sint Eustatius (Oranjestad Bay, 17.479 751°N −62.987 273°W). The culture experiments with *A. angulatus* and *S. marginalis* were conducted in the same manner as described in van Dijk et al.

(2017), with the exception of media preparation. Culture media of different salinities were prepared by mixing natural 0.2 μm filtered seawater with deionized water and 'instant ocean' salt, to obtain a range in salinities between 25–45. The *A. lessonii* specimens are from Burger's Zoo (van Dijk et al., 2019); the culture conditions are reported in van Dijk et al. (2019). The specimens of *A. tepida* were collected on a tidal flat near Den Oever, the Wadden Sea, NL (Hayward et al. 2004); the culture conditions are reported in Geerken et al. (2018).