# Peer review of "Distribution of chlorine and fluorine in benthic foraminifera"

_Biogeosciences, 2019_

## Short Comment (SC1) · 3 Dec 2019

Thank you for presenting for the first time data on the distribution of F and Cl in foraminiferal calcite. I have some short comments of issues I noticed during a quick read of the manuscript, which are mainly concerning the lack of details of the culture experiments and the graphical presentation of the data. I leave a proper review to the invited referees.

Fig 1. The miliolid species come from two salinity conditions, according to table 1. From which salinities are the specimens show in Fig. 1? And which chambers: ultimate, pen-ultimate, etc? I think a SEM picture of the studied areas would be a good addition to Fig. 1. I see the general overview pictures in the appendix, but I would like to see also

the higher magnification image.

Looking at the location of the measurements of the miliolids, and the explanation of the culture set-up, how can you assure the measurements were done on newly formed (experimental) calcite? Judging the orientation of the foraminifera in the SEM images in Appendix A1, it seems like you are not measuring e.g. the last chambers, which are a bit less complex. Especially in the case of Archaias, the last chambers seem to be on the top left of the image, and it looks likes the authors choose a quite complex location for the analysis. Why not analyse the last chambers, where the direction of growth is more clear? Also, the polishing of the Sorites doesn't seem to include the last chambers, because they appear to be still inside the resin (or was the specimen broken?). Please indicate the chamber numbers (F, F-1 etc) and, most important, which ones are precipitated in the experiment. This is crucial, since the authors compare to the culture conditions in Fig. 3 and Fig. A4. Also, please mirror the scalebar in these figures for readability.

In my opinion, the culture experiments have to be described more in detail, clearly stating the differences between the set-ups. Even though the other experiments are published already, some basic details can be stated in section 2.1. Also there is no clear indication how samples were cleaned, while the cleaning can have a major effect on the element distribution (Glock et al., 2019: https://doi.org/10.3389/feart.2019.00175). Digging through the publications of the other experiments, this information can be retrieved. But cleaning method is not presented for the unpublished experiment. Please add this information.

How were the E/Ca measured for the milliolid species? Since these specimens are coming from an unpublished experiment, and are not "previously described (Geerken et al., 2018; van Dijk et al., 2019)", as the authors state. Please give these details.

It looks like the Archaias angulatus cultured at salinity 40 has lower Na/Ca then the specimens from salinity of 30. What are the consequences for the Na/Ca – salinity

proxy, and the idea that milliolids are precipitating from seawater vacuoles?

Also indicate the salinity conditions in the figures/captions in figures and tables, e.g. appendix A1, and table A1.

Consider changing the terminology, from rotaliid to hyaline and milliolid to porcelaneous.

For future work, please also consider to analyse also the natural chambers from the field for comparison with the experimental chambers. Especially for the specimens that were culture using Instant ocean salt, which is an industrial manufactured salt, lacking e.g. certain organic complexes. Also, as indicated in van Dijk et al., 2019 (https://www.frontiersin.org/articles/10.3389/feart.2019.00281/full), there is a high intra- and inter-specimen variability for many elements. Therefore, please consider measuring several chambers and specimens to gain a robust dataset.

---

## Referee Comment (RC1) · Anonymous Referee #1 · 25 Mar 2020

Review for "Distribution of chlorine and fluorine in benthic foraminifera" by Roepert et al. This study looks at the incorporation of the anions chlorine and fluorine into the shells of benthic foraminifera. This has received almost no attention yet, although the conservative nature of these elements on the oceans would make them interesting to function as paleo proxies. Four different species, both rotaliid and miliolid, were cultured under controlled conditions. Analyses were performed using laser ablation and nanoSims. The distribution of chlorine and fluorine in the calcite varies between the low/intermediate-Mg species and the high-Mg species. Similar to other elements the lower Mg species show a clear banding of Cl and F related to the organic linings formed during biomineralization. As the biomineralization process is different in the high-Mg species in that no banding is developed, this is also not visible in the Cl and F

content. So, the distribution of Cl and F depends on the biomineralization process and seems mostly connected to organic content.

The manuscript is well written and organized, it is easy to read and extensive details on the methods are given. I do miss a few things on the methods though, and a final implications section or paragraph (see below). I recommend that this manuscript makes a valuable addition to Biogeosciences after minor revisions have been made.

In the abstract the potential of these conservative elements as paleo-proxy is mentioned, but then apart from one sentence (Line 182) this is not coming back anymore. I suggest to include a final paragraph at the end of the discussion what these results imply for proxy development. Is it possible at all to conclude something about this? It is stated already that the number of samples and different setups is not large enough to identify trends, but could the extremely high-resolution also be an issue to determine their use as a proxy? For a commonly used proxy as Mg/Ca you also see a very heterogeneous distribution when looking at the micro-scale that does not appear to correlate with environmental conditions. But the actual proxy is the ratio that is representative for the whole shell (or enough laser profiles). So, how representative do you think your results are? Just six specimens on four different species, and a laser profile through each one showing how heterogeneous the distributions are, is not very much.

Section 2.1: More details on the culturing experiments are needed. Part of them are in Appendix B, but I think this would be much better to include into the main text. What I miss is on what part of the forams the analyses were done. I assume on the newly grown calcite, but how was this determined? Did you use a marker in the solution, or simply took the last chamber? A comparison with the original, naturally-grown calcite would also be interesting. What were the concentrations of these elements in the culture solutions; similar to sea water? The saturation state of the angulatus and marginalis experiments is very high. Were there any indicators of inorganic precipitation of calcite, which could have biased the results?

---

## Referee Comment (RC2) · Anonymous Referee #2 · 15 Apr 2020

Roepert et al present NanoSIMS results looking at the distribution of chlorine and fluorine in cultured benthic foraminifera; two rotaliid species where calcite test walls are constructed via calcification around a primary organic sheet (hyaline calcification), and two miliolid species where test walls are constructed from calcite needles within an organic matrix.

The preliminary results show that the calcification pathway of benthic foraminifera determines the incorporation and distribution of Cl, F, P and other elements in their calcite shells.

The paper is interesting and well written and a good fit to Biogeosciences. One thing that is missing from the text relates to what kind of proxy the authors think the various halogen elements versus calcium ratios would represent?

I have a couple of minor comments that can easily be addressed with minor revisions:

- The study takes advantage of benthic foraminifera cultured for different purposes, under different conditions (Figure 4). Were all the samples cultured in the same artificial/natural seawater, and were halogen concentrations monitored? Several of the environmental parameters were calculated from other relationship (salinity-alkalinity) rather than measured. How constant would these parameters have been during the culture experiments? It would be good to see a discussion of error estimates relating to the parameters the halogen/Ca are being compared with. Furthermore, a brief discussion about halogen/Ca errors/variability also seems appropriate.

- All results are grouped together in Figure 3 and 4. Why would you expect a similar relationship between halogen/Ca and environmental parameters in hyaline and miliolid species?

- Correlations. Tone down discussion concerning correlations as only very few specimens were used of the same species etc in abstract and results section.

- Spatial distribution of halogen/Ca (Figure 1). For the hyaline species higher values are found in the primary organic sheet for all three halogens. Have the authors taken into consideration that Ca in the primary organic sheet will be much lower than in the calcite? Halogen/Ca ratios are hence higher, but it doesn't mean that halogens are actually higher in concentration than they are in the calcite. Do the anion counts show elevated concentrations in these bands?

Other comments:

Abstract: The discussion of the results is vague. What is meant by 'Cl and F were highly heterogeneous and correlated within the shell walls' (line 7, 8), and 'In these species Cl and P were correlated' (line 10)? was the correlation positive or negative, and how significant? Lines 14, 15 'We further propose that in the miliolid species Cl may be incorporated as a solid solution of chlorapatite or associated with organic molecules in

the calcite'. It is unclear what is meant with solid solution? Do you mean chloroapatite that has dissolved? Perhaps not use the word organic lining as a pseudonym for primary organic sheet, as foraminifera sometimes have an organic lining on the inside of the test.

4.1 How could you check if fluorite of fluorapatite are the incorporation mechanism for fluorine in calcite? Has there been a discussion about this with regards to aragonite which is also higher in F?

Figure 5 What is new here compared with previous work? Needs appropriate referencing.

Figure A1 SEM images are mirrored. Please change back!

---

## Author Comment (AC1) · 24 Jun 2020

Short Comment 1

Thank you for presenting for the first time data on the distribution of F and Cl in foraminiferal calcite. I have some short comments of issues I noticed during a quick read of the manuscript, which are mainly concerning the lack of details of the culture experiments and the graphical presentation of the data. I leave a proper review to the invited referees.

Thank you for the feedback. We will provide further details on the culture experiments and materials used.

Comment SC1.1: Fig 1. The miliolid species come from two salinity conditions, ac-

cording to table 1. From which salinities are the specimens show in Fig. 1? And which chambers: ultimate, penultimate, etc?

Answer: The salinities for the specimens shown in Figure 1 were presented in Table 1. As this study does not focus on salinity, we decided not to report salinities (and other environmental parameters) in the figure or figure caption, but instead to provide an overview in Table 1. We agree that the information in Table 1 could only be linked to Figure 1 via Fig. A1, which was suboptimal.

Changes: To facilitate identification, we added specimen numbers to Figure 1 and we explicitly mention in the figure caption that details on the specimens used are presented in Table 1. Chamber numbers (F, F-1, etc.) were added to Fig. A1.

Comment SC1.2: I think a SEM picture of the studied areas would be a good addition to Fig. 1. I see the general overview pictures in the appendix, but I would like to see also the higher magnification image.

Answer: Figure 1 is already rather complex and we rather refrain from adding even more complexity to it. The context of the detailed nanoSIMS images are provided by SEM images shown in Fig. A1. These are high resolution images that can be zoomed in by the reader.

Comment SC1.3: Looking at the location of the measurements of the miliolids, and the explanation of the culture set-up, how can you assure the measurements were done on newly formed (experimental) calcite? Judging the orientation of the foraminifera in the SEM images in Appendix A1, it seems like you are not measuring e.g. the last chambers, which are a bit less complex. Especially in the case of Archaias, the last chambers seem to be on the top left of the image, and it looks likes the authors choose a quite complex location for the analysis. Why not analyse the last chambers, where the direction of growth is more clear? Also, the polishing of the Sorites doesn't seem to include the last chambers, because they appear to be still inside the resin (or was the specimen broken?).

[Figure]

Answer: The cultures were started with juvenile specimens, possessing 2-3 chambers at the start of the experiment. All additional chambers were formed during the course of the experiment. The miliolid species were cultured in media containing the fluorescent indicator calcein to identify newly formed calcite in retrospect. Positions for nanoSIMS imaging were carefully selected based on the quality of the surface preparation and position in the specimen. Where possible, distal chambers have been measured, but more proximal chambers were preferred in case their cross-sectional surfaces were flatter or cleaner.

Changes: The following information was added to the methods section: "The fields of view for NanoSIMS imaging were carefully selected using SEM images on the basis of the position in the specimen and the quality of the surface preparation. Where possible, distal chambers were measured, but more proximal chambers were preferred if their cross-sectional surfaces appeared flatter or cleaner."

Comment SC1.4: Please indicate the chamber numbers (F, F-1 etc) and, most important, which ones are precipitated in the experiment. This is crucial, since the authors compare to the culture conditions in Fig. 3 and Fig. A4.

Answer: This comment relates to SC1.3, see also the answer to SC1.3. We assume this comment refers to the rotaliid species, where chambers are commonly indicated with F, F-1, etc.

Changes: We indicated chamber numbers in Fig. A1 for the rotaliid specimens.

Comment SC1.5: Also, please mirror the scalebar in these figures for readability.

Answer: Done.

Changes: Scale-bar text mirrored for better readability.

Comment SC1.6: In my opinion, the culture experiments have to be described more in detail, clearly stating the differences between the set-ups. Even though the other experiments are published already, some basic details can be stated in section 2.1. Also there is no clear indication how samples were cleaned, while the cleaning can have a major effect on the element distribution (Glock et al., 2019: https://doi.org/10.3389/feart.2019.00175). Digging through the publications of the other experiments, this information can be retrieved. But cleaning method is not presented for the unpublished experiment. Please add this information.

Answer: Done.

Changes: The following was added to the methods section: "The A. angulatus and S. marginalis specimens were collected in Sint Eustatius (Oranjestad Bay, 17.479751°N -62.987273°W). The culture experiments with A. angulatus and S. marginalis were conducted in the same manner as described in van Dijk et al. (2017), with the exception of media preparation. Culture media of different salinities were prepared by mixing natural 0.2$\mu$m filtered seawater with deionized water and 'instant ocean' salt, to obtain a range in salinities between 25-45. Calcein was added during the course of the experiment, and fluorescence images were used to identify newly precipitated calcite. The A. lessonii specimens are from Burger's Zoo, NL (van Dijk et al., 2019), with the culture conditions being reported in van Dijk et al. (2019). The specimens of A. tepida were collected on a tidal flat near Den Oever, the Wadden Sea, NL (Hayward et al., 2004), with the culture conditions being described in Geerken et al. (2018). For both the cultures of A. lessonii and A. tepida, 2-3 chambered juveniles were transferred into Petri dishes containing culture media with adjusted salinity and alkalinity, where the specimen precipitated additional chambers. Prior to embedding all specimens were cleaned using an adapted Barker protocol (Barker et al., 2013), only applying the organic removal/oxidation step, in which NaOH was replaced by NH4OH, as described in detail in Geerken et al. (2018)."

Comment SC1.7: How were the E/Ca measured for the milliolid species? Since these specimens are coming from an unpublished experiment, and are not "previously described (Geerken et al., 2018; van Dijk et al., 2019)", as the authors state. Please give these details.

Answer: We meant the El/Ca ratios in the miliolid species were determined using the same methodology "as previously described [...]".

Changes: for more clarity the text was adapted to "This was done by LA-ICP-MS for A. tepida and A. lessonii as previously described (Geerken et al., 2018; van Dijk et al., 2019). For A. angulatus and S. marginalis LA-ICP-MS analyses were performed using the same methodology as described in Geerken et al. (2018)."

Comment SC1.8: It looks like the Archaias angulatus cultured at salinity 40 has lower Na/Ca then the specimens from salinity of 30. What are the consequences for the Na/Ca – salinity proxy, and the idea that milliolids are precipitating from seawater vacuoles?

Answer: Large intra-specimen variability in Na/Ca has been shown for rotaliid species (e.g. Geerken et al., 2018). It may well be that miliolids exhibit even large intra-specimen Na/Ca variability as well, where a specimen cultured at salinity 40 can have a lower Na/Ca than a specimen cultured at a salinity of 30. To be able to draw any conclusions on what the consequences of individual specimen Na/Ca would be for the Na/Ca salinity proxy using miliolid species, further research is needed. This should involve culturing experiments using a statistically sound number of replicate specimen at a range of salinities.

Changes: As this manuscript focuses on the anions Cl and F, we have not included a note about the range in Na/Ca of the presented specimens.

Comment SC1.9: Also indicate the salinity conditions in the figures/captions in figures and tables, e.g. appendix A1, and table A1. Answer: See answer to comment SC1.1.

Changes: We have added the specimen number where missing to facilitate finding the respective environmental conditions in Table 1.

Comment SC1.10: Consider changing the terminology, from rotaliid to hyaline and milliolid to porcelaneous.

Answer: We have considered different terminology, but chose for referring to the differences in terms of order instead of test appearance. We did so, because hyaline foraminifera also include globigerinids, which were not investigated in this study.

Comment SC1.11: For future work, please also consider to analyse also the natural chambers from the field for comparison with the experimental chambers. Especially for the specimens that were culture using Instant ocean salt, which is an industrial manufactured salt, lacking e.g. certain organic complexes.

Answer: a valuable suggestion for future work.

Comment SC1.12: Also, as indicated in van Dijk et al., 2019 (https://www.frontiersin.org/articles/10.3389/feart.2019.00281/full), there is a high intra- and inter-specimen variability for many elements. Therefore, please consider measuring several chambers and specimens to gain a robust dataset.

Answer: we are aware that a more robust data set is needed for drawing conclusions concerning proxy potential and relationships with environmental parameters. However, as we here present a pilot study into the spatial distribution of Cl and F in rotaliid vs. miliolid benthic foraminiferal species, we regard the current data set sufficient. We agree that future research using more replicates and consistent culturing conditions is needed to better understand the incorporation mechanisms and impact of environmental conditions on incorporation of Cl and F.

Changes: see also comment RC1.1. In our revised version, we have stated more clearly that the current data set does not allow for conclusions on proxy application.

---

## Author Comment (AC2) · 24 Jun 2020

Referee Comments Anonymous Referee #1

Review for "Distribution of chlorine and fluorine in benthic foraminifera" by Roepert et al. This study looks at the incorporation of the anions chlorine and fluorine into the shells of benthic foraminifera. This has received almost no attention yet, although the conservative nature of these elements on the oceans would make them interesting to function as paleo proxies. Four different species, both rotaliid and miliolid, were cultured under controlled conditions. Analyses were performed using laser ablation and nanoSims. The distribution of chlorine and fluorine in the calcite varies between the low/intermediate-Mg species and the high-Mg species. Similar to other elements

the lower Mg species show a clear banding of Cl and F related to the organic linings formed during biomineralization. As the biomineralization process is different in the high-Mg species in that no banding is developed, this is also not visible in the Cl and F content. So, the distribution of Cl and F depends on the biomineralization process and seems mostly connected to organic content. The manuscript is well written and organized, it is easy to read and extensive details on the methods are given. I do miss a few things on the methods though, and a final implications section or paragraph (see below). I recommend that this manuscript makes a valuable addition to Biogeosciences after minor revisions have been made.

We thank the referee for this constructive feedback. Details of our response are given below.

Comment RC1.1: In the abstract the potential of these conservative elements as paleo-proxy is mentioned, but then apart from one sentence (Line 182) this is not coming back anymore. I suggest to include a final paragraph at the end of the discussion what these results imply for proxy development. Is it possible at all to conclude something about this? It is stated already that the number of samples and different setups is not large enough to identify trends, but could the extremely high-resolution also be an issue to determine their use as a proxy?

Answer: Our study did not aim at the development of a new proxy, but rather at exploring the incorporation of halogens. Consequently the data obtained do not allow to draw conclusions for proxy development. A more robust data set with, species-specific, replicated specimen per treatment would be needed. Also, large intra-test variability that is observed for many trace elements in foraminiferal shells using high-resolution imaging techniques implies replicate analyses on several chambers per specimen are necessary. Creating large data sets with replicate measurements on many specimens is not the strength of NanoSIMS and hence other analytical techniques are more suitable for studying potential proxy applicability. Still, our approach does provide distributional data of F and Cl so far not available.
Changes: We added a clearer statement to the discussion that potential proxy applicability of Cl/Ca and F/Ca cannot be evaluated based on our data.

Comment RC1.2: For a commonly used proxy as Mg/Ca you also see a very heterogeneous distribution when looking at the micro-scale that does not appear to correlate with environmental conditions. But the actual proxy is the ratio that is representative for the whole shell (or enough laser profiles). So, how representative do you think your results are? Just six specimens on four different species, and a laser profile through each one showing how heterogeneous the distributions are, is not very much.

Answer: We acknowledge that our data set is limited to few fields of view, on a limited amount of specimens. However, lateral profiles have also been made for the fields of view where the images are not shown. The panels in Figure 1 show a representative image per species of those we analysed, and based on the similarity between the images within one species we do not expect appreciable differences in elemental distribution patterns if we imaged more fields of view. For the purpose of presenting key differences between rotaliid and miliolid species we consider our data sufficient. The data that are shown in Figures 3, 4 and A4 are averages of 1, 2 or 3 fields of view per specimen including standard deviation. They resemble average elemental ratios as determined by LA-ICP-MS measurements, although, at a higher resolution. Since (in the case of the rotaliids) the field of view of the NanoSIMS images covered a cross-section through the shell wall, we expect the average of one NanoSIMS image to resemble an average LA-ICP-MS profile.

Changes: We have added "As such, lateral profiles that cover a representative fraction of a shell wall may be comparable to LA-ICP-MS profiles, albeit with a higher resolution." to the methods section.

Comment RC1.3: Section 2.1: More details on the culturing experiments are needed. Part of them are in Appendix B, but I think this would be much better to include into the main text.

Answer: This comments echoes that of Inge van Dijk and for a detailed response see answer to Comment SC1.6.

Changes: more details have been added in the main text, see answer to Comment SC1.6.

Comment RC1.4: What I miss is on what part of the forams the analyses were done. I assume on the newly grown calcite, but how was this determined? Did you use a marker in the solution, or simply took the last chamber?

Answer: This comment relates to Comment SC1.3. Please refer to SC1.3 for a detailed answer.

Changes: See Comment SC1.3.

Comment RC1.5: A comparison with the original, naturally-grown calcite would also be interesting.

Answer: A valuable suggestion for future research, but for the scope of this pilot study we consider the current data set sufficient.

Comment RC1.6: What were the concentrations of these elements in the culture solutions; similar to sea water?

Answer: We did not determine the concentrations of Cl and F in the culture media directly. Since Cl and F are conservative elements following salinity, the concentrations are expected to resemble those in sea water with the same salinity.

Changes: The text has been modified to explicitly mention this: "The concentrations of Cl and F in the culture media were not directly determined. However, since Cl and F are conservative elements following salinity, the concentrations are expected to resemble those in sea water with the same salinity."

Comment RC1.7: The saturation state of the angulatus and marginalis experiments is very high. Were there any indicators of inorganic precipitation of calcite, which could

have biased the results?

Answer: During the experiments there were no visual indicators of inorganic precipitation of calcite. The obtained specimens did not show visual overgrowth under the SEM.

Changes: We have added the following: "During the culture experiments there were no visual indicators of inorganic precipitation of calcite. Moreover, inspection with SEM of the measured specimens showed no inorganic calcite overgrowth."

———————————————————

---

## Author Comment (AC3) · 24 Jun 2020

Referee Comments Anonymous Referee #2

Roepert et al present NanoSIMS results looking at the distribution of chlorine and fluorine in cultured benthic foraminifera; two rotaliid species where calcite test walls are constructed via calcification around a primary organic sheet (hyaline calcification), and two miliolid species where test walls are constructed from calcite needles within an organic matrix. The preliminary results show that the calcification pathway of benthic foraminifera determines the incorporation and distribution of Cl, F, P and other elements in their calcite shells. The paper is interesting and well written and a good fit to Biogeosciences. One thing that is missing from the text relates to what kind of proxy

the authors think the various halogen elements versus calcium ratios would represent? I have a couple of minor comments that can easily be addressed with minor revisions:

Thank you for your constructive feedback: our detailed responses are given below.

Comment RC2.0: One thing that is missing from the text relates to what kind of proxy the authors think the various halogen elements versus calcium ratios would represent?

Answer: This information is given in the introduction in lines 42-44.

Comment RC2.1: The study takes advantage of benthic foraminifera cultured for different purposes, under different conditions (Figure 4). Were all the samples cultured in the same artificial/natural seawater, and were halogen concentrations monitored?

Answer: The specimen were not cultured in the same artificial/natural seawater. However, the range of salinities created in the culture media by modifications of the natural and artificial seawater based culture media were larger than expected differences between natural and artificial seawater at the same salinity. For the experiments with A. lessonii and A. tepida, the culture media were produced from natural seawater, while for the culture experiments of S. marginalis and A. angulatus, artificial seawater based culture media were used. As halogen concentrations in seawater are tightly linked with salinity, we chose to determine, for practical reasons, salinity in the culture media stocks rather than halogen concentrations. See also answer to comment RC1.6.

Changes: More detail on culturing is provided in the methods section in the revised version: for the detailed changes, see comment RC1.6.

Comment RC2.2: Several of the environmental parameters were calculated from other relationship (salinity-alkalinity) rather than measured. How constant would these parameters have been during the culture experiments?

Answer: As it was not feasible to measure these parameters in the small Petri dishes the foraminifera were grown in, the parameters were measured once when preparing the seawater stock. During the experiment, these variables may have varied slightly

due to evaporation when feeding/cleaning. However, the media were replaced with fresh stock seawaters with the fixed parameters twice a week. Furthermore, culture media were in equilibrium with atmospheric pCO2. Changes in DIC due to calcification are expected to be negligible given the high ratio of culture media volume to foraminiferal calcite per Petri dish.

Changes: We have added more information on the culturing to the Methods section: "Due to the small culturing volumes (Petri dishes), the parameters of the media could not be monitored during the experiments. However, potential changes due to evaporation during feeding or cleaning of the cultures are expected to be negligible, because the culture media were renewed regularly (twice a week), when compared to the large differences between the treatments."

Comment RC2.3: It would be good to see a discussion of error estimates relating to the parameters the halogen/Ca are being compared with.

Answer: see also comment RC2.2. The ranges of the environmental parameters to which the halogen/Ca ratios are compared to, are in the extremes of what is found in natural seawaters, e.g. total alkalinity ranged from 1350 – 4477 $\mu$mol/kg. We therefore expected that slight variations in the culture media had little impact with respect to the large differences between the culture media.

Changes: We have added more information on the culturing to the Methods section.

Comment RC2.4: Furthermore, a brief discussion about halogen/Ca errors/variability also seems appropriate.

Answer: see Comment RC1.2.

Changes: see Comment RC1.2.

Comment RC2.5: All results are grouped together in Figure 3 and 4. Why would you expect a similar relationship between halogen/Ca and environmental parameters in hyaline and miliolid species?

Answer: Since this is the first detailed study of Cl and F incorporation into benthic foraminiferal calcite, we presented the data both in terms of co-variance with cations that have been investigated in detail before and in terms of carbonate system parameters. We made no a priori assumption about the absence or presence of a relationship. When halogen incorporation would not be dominated by biomineralization pathway differences of rotaliid and miliolid species, but governed (mainly) by environmental parameters, then a relationship may be expected, as discussed in lines 199-200.

Comment RC2.6: Correlations. Tone down discussion concerning correlations as only very few specimens were used of the same species etc in abstract and results section.

Answer: This comment echoes RC1.1. We are aware that the number of specimens measured in this study does not allow a robust interpretation of the effects of environmental parameters. We have nevertheless included figures showing our data in comparison to culture media properties for visualization.

Changes: see changes to comment RC1.1.

Comment RC2.7: Spatial distribution of halogen/Ca (Figure 1). For the hyaline species higher values are found in the primary organic sheet for all three halogens. Have the authors taken into consideration that Ca in the primary organic sheet will be much lower than in the calcite? Halogen/Ca ratios are hence higher, but it doesn't mean that halogens are actually higher in concentration than they are in the calcite. Do the anion counts show elevated concentrations in these bands?

Answers: Yes, we have taken into consideration that Ca intensities may be lower in the locations of the organic sheet compared to the calcite. However, due to the spatial resolution this is hardly visible in the Ca intensities and the anion intensities are elevated in the locations of the bands, see added figure A6.

Changes: We have added Figure A6 to the appendix showing the elemental intensity profiles for the same transects as in Figure 1, illustrating that elevated halogen/Ca

ratios at the locations of organic linings are not caused by lower Ca intensities. We furthermore added "These bands are not caused by lower Ca intensities at these locations (Figure A6)." to the Results section.

Other comments: Comment RC2.8: Abstract: The discussion of the results is vague. What is meant by 'Cl and F were highly heterogeneous and correlated within the shell walls' (line 7, 8), and 'In these species Cl and P were correlated' (line 10)? was the correlation positive or negative, and how significant?

Answer: 'Cl and F were highly heterogeneous and correlated within the shell walls' (line 7, 8) refers to the spatial distribution of Cl and F within shell walls. 'In these species Cl and P were correlated' (line 10) refers to a positive spatial correlation of Cl and P.

Changes: the text was adapted as follows: "Cl and F were distributed highly heterogeneously within the shell walls, forming bands that were co-located with the bands observed in the distribution of phosphorus (significant positive correlation of both Cl and F with P; $p < 0.001$)" and "In these species Cl and P were spatially positively correlated ($p < 0.001$)"

Comment RC2.9: Lines 14, 15 'We further propose that in the miliolid species Cl may be incorporated as a solid solution of chlorapatite or associated with organic molecules in the calcite'. It is unclear what is meant with solid solution? Do you mean chloroapatite that has dissolved?

Answer: The term "solid solution" is a standard term used in thermodynamics and mineralogy for mixtures of solid phases that have similar crystal structures (similar to aqueous solutions in aquatic environments).

Comment RC2.10: Perhaps not use the word organic lining as a pseudonym for primary organic sheet, as foraminifera sometimes have an organic lining on the inside of the test.

Answer: We acknowledge that foraminifera can have organic linings additional to the

primary organic sheet.

Changes: We added "Here, we use the term organic linings to refer collectively to the primary organic sheet and other organic linings in the shell wall." to the discussion for clarity.

Comment RC2.11: 4.1 How could you check if fluorite of fluorapatite are the incorporation mechanism for fluorine in calcite? Has there been a discussion about this with regards to aragonite which is also higher in F?

Answer: Spectroscopic techniques such as synchrotron could potentially identify incorporation mode of F in foraminiferal calcite, hence, whether fluorite or fluorapatite may play a role here. This requires further investigations outside the scope of this study. The incorporation mechanism for F into aragonite can be attributed to ion exchange with carbonate ion (Ichikuni, Chemical Geology 1979) and to the best of our knowledge alternatives have not been discussed yet.

Changes: We have added text to the discussion to point towards future research options.

Comment RC2.12: Figure 5 What is new here compared with previous work? Needs appropriate referencing.

Answer: done.

Changes: References were added to the figure caption.

Comment RC2.13: Figure A1 SEM images are mirrored. Please change back!

Answer: The SEM images in Figure A1 are mirrored on purpose to represent the orientation of the nanoSIMS images. Mirroring them back would complicate visual comparison of the SEM images and nanoSIMS images. We therefore decided to leave the SEM images in their mirrored state.

Changes: we have flipped back the scale bar annotation for better readability and

added "SEM images are flipped horizontally to facilitate navigation in the NanoSIMS instrument, where the secondary ion images are horizontally mirrored." to the caption of Figure A1.